# Cost-effectiveness of sleeping sickness elimination campaigns in five settings of the Democratic Republic of Congo

Marina Antillon [1,2✉], Ching-I Huang [3,4], Ronald E. Crump [3,4], Paul E. Brown[3,4], Rian Snijders [1,2,5], Erick Mwamba Miaka[6], Matt J. Keeling[3,4,7], Kat S. Rock [3,4,8✉] & Fabrizio Tediosi [1,2,8]

Gambiense human African trypanosomiasis (gHAT) is marked for elimination of transmission by 2030, but the disease persists in several low-income countries. We couple transmission and health outcomes models to examine the cost-effectiveness of four gHAT elimination strategies in five settings – spanning low- to high-risk – of the Democratic Republic of Congo. Alongside passive screening in fixed health facilities, the strategies include active screening at average or intensified coverage levels, alone or with vector control with a scale-back algorithm when no cases are reported for three consecutive years. In high or moderate-risk settings, costs of gHAT strategies are primarily driven by active screening and, if used, vector control. Due to the cessation of active screening and vector control, most investments (75-80%) are made by 2030 and vector control might be cost-saving while ensuring elimination of transmission. In low-risk settings, costs are driven by passive screening, and minimum-cost strategies consisting of active screening and passive screening lead to elimination of transmission by 2030 with high probability.

[1] Epidemiology and Public Health, Swiss Tropical and Public Health Institute, Allschwil, Basel 4123, Switzerland. [2] University of Basel, Basel 4001, Switzerland. [3] Zeeman Institute, University of Warwick, Coventry CV4 7AL, UK. [4] Mathematics Institute, University of Warwick, Coventry CV4 7AL, UK. [5] Institute of Tropical Medicine, B-2000 Antwerp, Belgium. [6] Programme National de Lutte contre la Trypanosomiase Humaine Africaine (PNLTHA), Kinshasa, Democratic Republic of Congo. [7] School of Life Sciences, University of Warwick, Coventry CV4 7AL, UK. [8] These authors contributed equally: Kat S. Rock, Fabrizio Tediosi. ✉email: marina.antillon@swisstph.ch; k.s.rock@warwick.ac.uk

*Gambiense* human African trypanosomiasis (gHAT) has caused three decades-long epidemics in Western, Central, and Eastern-Sub-Saharan Africa during the twentieth century[1]. Caused by the parasite *Trypanosoma brucei gambiense* and transmitted by tsetse, the disease is nearly always fatal if untreated[2]. No vaccine exists against gHAT, and current treatments are incompatible with mass drug administration strategies, but there is a relatively diverse toolbox available to control the infection[3]: a range of diagnostics for screening in at-risk villages by mobile teams, termed as active screening (AS), as well as diagnostics to test symptomatic individuals in fixed health facilities, termed as passive screening (PS). A new oral therapeutic —fexinidazole—became available in 2020, expanding the treatments available for outpatient care[4]. In addition, vector control (VC) can now be deployed across moderate scales to reduce tsetse density[5–9]. This toolbox has enabled a vast reduction in burden from 37,000 reported cases at the peak of the epidemic in 1998–864 reported cases in 2019[10–12].

In 2012, under 15 years after the peak of the last epidemic, gHAT was targeted for elimination of transmission (EOT) by 2030, and by 2014 case reduction outpaced the intermediate goals set by the World Health Organization (WHO)[3,13]. Consequently, the pursuit of EOT brings new questions to the fore: because gHAT activities take place in settings facing both resource constraints and competing health needs, what are the resource implications of further pursuing gHAT EOT by 2030? Which are the most efficient strategies toward gHAT suppression?

The key economic arguments for and against disease elimination programs may appear at odds: (i) the long-term benefits of elimination or eradication and the subsequent savings borne out of scaled-back disease control measures are substantial, yet (ii) the endgame is comprised of increasingly more expensive activities on a per-case basis without the guarantee of success. Smallpox eradication is heralded as an example of (i): eradication activities saved $1.35 billion (US$) but had a cost of $100 million[14]. Whereas polio—for which there were 42 cases reported in 2016—exemplifies (ii), costing $3.3 billion each year, or $1 billion more than a suitable control strategy without eradication goal[15].

To date, there has been one cost-effectiveness analysis of control and elimination strategies for gHAT, showing that new technologies to facilitate diagnosis, treatment, and curtail vector transmission could make elimination feasible at a moderate cost-effectiveness ($400–1500 per disability-adjusted life year (DALY) averted) for high- and medium-transmission settings, but at low cost-effectiveness for low-transmission settings[16]. However, it is worth reconsidering these questions while taking into account specific, local transmission dynamics, which is now possible thanks to recent developments in model calibration[17], and new cost estimates[18,19]. Moreover, including realistic levels of screening based on regional data allows us to consider expenses and cost-effectiveness in these real-world settings.

In the current study, we undertake an economic evaluation of four gHAT control and elimination strategies in distinct transmission settings in five health zones of the Democratic Republic of Congo (DRC), the country that reported 74% of global cases in 2018[11] and 71% of global cases in 2019[20]. We adopt a modeling framework in order to examine the cost-effectiveness of gHAT elimination strategies while taking into account the short- and long-term interplay of epidemiological and economic factors of the disease.

## Results

We selected five health zones in DRC, described in Table 1, spanning the spectrum of the WHO's incidence categories[3]. For each health zone, we simulated four strategies, depicted in Fig. 1, using a variant of the "Warwick gHAT model", an SEIRS-type deterministic dynamic model that explicitly simulates the transmission of the disease between humans via tsetse (see Supplementary Methods, section A.2.1). The model was fitted to case data from the health zones, adjusting for both the screening coverage as well as the positivity rate, and projections for reported cases, unreported illnesses, and reported and unreported deaths were simulated for 2020–2050[17,21]. The same underlying model framework has also been used to examine the epidemiology of gHAT in DRC and Chad[6,17,22–25].

Strategies are configurations of the following interventions as delineated by the national policy and in line with WHO recommendations (Fig. 1 and Supplementary Methods, section A.2.2)[3,26]. Alongside passive screening (PS) in fixed health facilities, the strategies included active screening (AS) at (i) average or (ii) high-coverage levels equal to the historical maximum screening coverage, alone (i–ii) and in tandem with vector control (VC; iii–iv). Status quo strategies, or the current practice, are considered to be equivalent to Strategy 1, except in Yasa Bonga, where VC has been in place since mid-2015, and therefore we consider the status quo equivalent to Strategy 3 and we omit strategies without VC for Yasa Bonga. The strategies with historical maximum screening coverage allow us to compare the outcomes in all health zones when high amounts of resources are invested while still taking into account that maximum capacity might vary across locations. Treatment is modeled by a branching (probability tree) model simulating WHO-recommended treatment, which consists of fexinidazole on an inpatient or outpatient basis, and pentamidine or nifurtimox-eflornithine combination therapy (NECT) when fexinidazole is contraindicated based on patient characteristics (see Supplementary Methods, section A.2.3 for a breakdown of treatment groups)[4].

Additionally, the model has a transition from the suppression to post-elimination phase, devised to simulate cessation of AS and VC when no cases are reported for three consecutive years, allowing as well for re-activation of AS should a case be detected in a fixed health facility (i.e., through PS). Therefore, by relying on a realistic index of disease suppression—detected cases—our model shows not only whether strategies lead to earlier or later cessation of activities, but also whether strategies lead to mistakenly early cessation if screening activities are insufficient.

**Feasibility of gHAT elimination and health impact**. The feasibility of EOT and cessation of AS and VC are shown in Table 2. The probability of elimination was calculated as the proportion of the 10,000 model iterations where there were no new infections (detected or not) by 2030, and the probability of reactive screening (RS) was calculated as the proportion of the 10,000 model iterations where there was at least one case detected by PS after AS had been ended (trace plots in Figs. 2 and 3 show stable estimates can be reached with 10,000 iterations). The year that AS ends is the last year of AS activities before cessation; RS may occur after.

While the risk category of each health zone (Table 1) influences the year when EOT is expected—places with higher incidence likely meet EOT later than places with lower incidence—the implementation of VC is predicted to substantially expedite EOT across all moderate- and high-risk settings and lower the probability of RS operations.

In low-risk Budjala, EOT appears imminent with any strategy, although employing Max AS over Mean AS would marginally reduce the uncertainty. In the moderate-risk settings of Mosango and Boma Bungu, EOT may occur by 2030 even with only Mean AS (79% and >99% probability, respectively), and EOT is almost

**Table 1 Descriptive summaries of five health zones.**

| Characteristic | Yasa Bonga | Mosango | Kwamouth | Boma Bungu | Budjala |
|---|---|---|---|---|---|
| Former province (new province) | Bandundu (Kwilu) | Bandundu (Kwilu) | Bandundu (Mai-Ndombe) | Bas-Congo (Kongo Central) | Equateur (Sud-Ubangi) |
| Population (2016 est.) | 221,917 | 125,076 | 131,022 | 85,960 | 133,425 |
| Area (km$^2$) | 2606 | 2673 | 14,589 | 2866 | 4397 |
| Active screening as a percent of 2016 population (mean, max) | 57, 91 | 34, 60 | 48, 69 | 7.2, 29 | 0.41, 36 |
| gHAT testing centers (2014 est.) | 4 | 1 | 5 | 2 | 2 |
| Yearly incidence per 10,000 (2012–2016) | 4.87 | 2.19 | 16.79 | 1.37 | 0.05 |
| WHO Incidence category | Moderate | Moderate | High | Moderate | Very low |
| Vector control extent (linear km) | 210 | 100 | 432 | 100 | 100 |
| Vector control density (targets per linear km) | 60 | 40 | 20 | 40 | 40 |

For Yasa Bonga and Kwamouth, the amount of vector control performed was informed by current and planned practice. For Mosango, Boma Bungu, and Budjala, assumptions regarding vector control extent and intensity were based on the experience in places of similar incidence. Sensitivity analyses regarding the assumptions around vector control are found in the supplement and in the companion website.

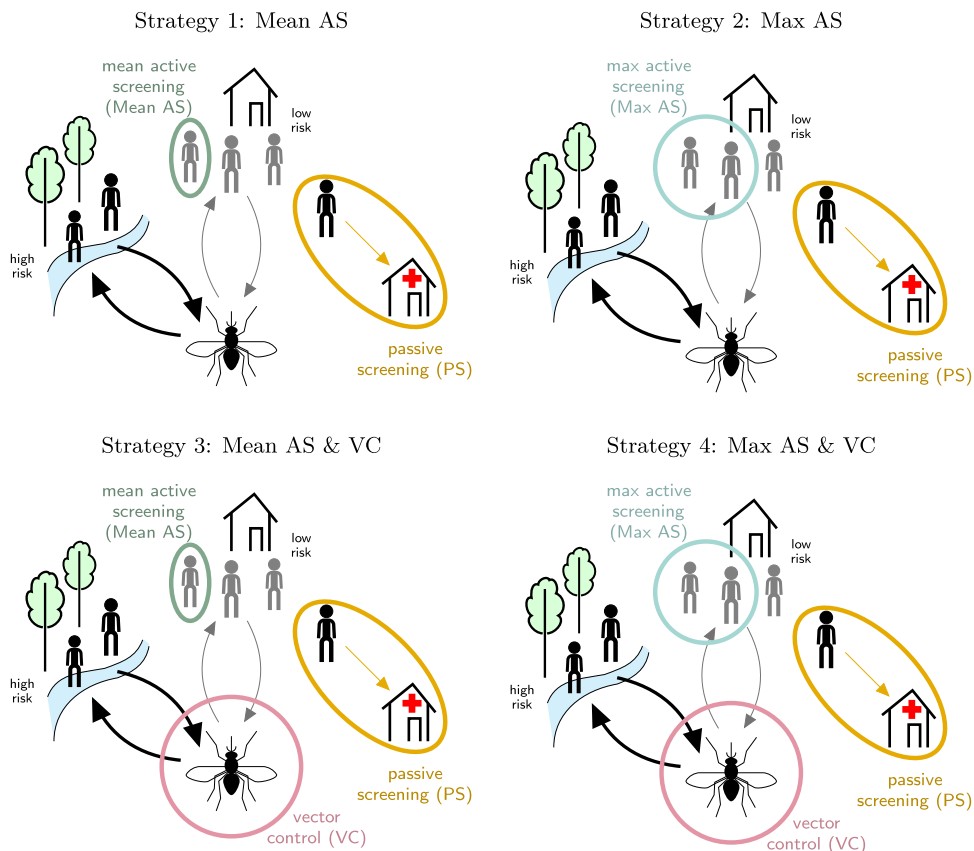

**Fig. 1 Model of strategies against gHAT in the Democratic Republic of Congo (DRC).** Strategies against gHAT, including active screening (AS) by mobile teams, passive screening (PS) in fixed health facilities, and vector control (VC). In two strategies ("Mean AS" and "Mean AS & VC") the proportion screened equalled the mean number screened during 2014–2018. In two other strategies ("Max AS" and "Max AS & VC"), the coverage is the maximum number screened in the period of 2000–2018. In strategies 3 and 4, VC is simulated assuming an 80% tsetse density reduction in one year, except in Yasa Bonga, where the reduction of tsetse density was estimated at 90% in the literature[9]. PS is in place under all strategies. This figure has been adapted from[24] under a CC-BY license.

certain by 2040. With additional AS coverage, the predicted probability of EOT by 2030 increases from 79% to 92% in Mosango, but VC would both ensure EOT by 2030 and bring forward the mean time to elimination by five years. In Yasa Bonga, where VC activities began in 2015 and scaled up completely by 2017, the model predicts that EOT may have been achieved by 2017 if VC was present in all areas of ongoing

transmission, although predicted case detections are still expected in the early 2020s followed by AS cessation in 2024 (95% CI: 2021–2028). In high-risk Kwamouth, EOT is predicted to be almost impossible by 2030 (<0.01) in the absence of VC, and unlikely by 2040 (11–13% depending on AS coverage), but adding VC is predicted to bring forward EOT by more than two decades. Zero detections are more informative as an EOT proxy when VC

**Table 2 Feasibility of elimination (additional scenarios are shown in the supplement).**

|  | Year of EOT (95% PI) | Prob. EOT by 2030 | Prob. EOT by 2040 | Year AS ends (95% PI) | Prob. RS |
|---|---|---|---|---|---|
| **Yasa Bonga** |  |  |  |  |  |
| Mean AS & VC | 2017 (2016, 2017) | >0.99 | >0.99 | 2024 (2021, 2028) | 0.11 |
| Max AS & VC | 2017 (2016, 2017) | >0.99 | >0.99 | 2024 (2021, 2028) | 0.12 |
| **Mosango** |  |  |  |  |  |
| Mean AS | 2028 (2021, 2037) | 0.79 | 0.99 | 2028 (2022, 2036) | 0.39 |
| Max AS | 2026 (2021, 2033) | 0.92 | >0.99 | 2027 (2022, 2033) | 0.33 |
| Mean AS & VC | 2021 (2020, 2021) | >0.99 | >0.99 | 2025 (2022, 2028) | 0.09 |
| Max AS & VC | 2021 (2020, 2021) | >0.99 | >0.99 | 2025 (2022, 2028) | 0.07 |
| **Kwamouth** |  |  |  |  |  |
| Mean AS | 2048 (2036, Post-2050) | <0.01 | 0.11 | 2043 (2034, Post-2050) | 0.58 |
| Max AS | 2047 (2036, Post-2050) | <0.01 | 0.13 | 2043 (2033, Post-2050) | 0.62 |
| Mean AS & VC | 2022 (2022, 2023) | >0.99 | >0.99 | 2029 (2026, 2035) | 0.12 |
| Max AS & VC | 2022 (2022, 2023) | >0.99 | >0.99 | 2029 (2026, 2035) | 0.13 |
| **Boma Bungu** |  |  |  |  |  |
| Mean AS | 2019 (2017, 2022) | >0.99 | >0.99 | 2023 (2021, 2027) | 0.02 |
| Max AS | 2019 (2017, 2022) | >0.99 | >0.99 | 2023 (2021, 2026) | 0.02 |
| Mean AS & VC | 2018 (2017, 2020) | >0.99 | >0.99 | 2022 (2021, 2026) | 0.02 |
| Max AS & VC | 2018 (2017, 2020) | >0.99 | >0.99 | 2022 (2021, 2025) | 0.01 |
| **Budjala** |  |  |  |  |  |
| Mean AS | 2023 (2017, 2031) | 0.97 | >0.99 | 2023 (2020, 2030) | 0.36 |
| Max AS | 2021 (2017, 2024) | >0.99 | >0.99 | 2023 (2020, 2027) | 0.22 |
| Mean AS & VC | 2020 (2017, 2024) | >0.99 | >0.99 | 2023 (2020, 2026) | 0.19 |
| Max AS & VC | 2020 (2017, 2023) | >0.99 | >0.99 | 2023 (2020, 2026) | 0.15 |

Estimates shown are means and their 95% prediction intervals (PI). Prob. EOT (elimination of transmission) is calculated as a proportion of the iterations of the dynamic transmission model for which transmission has reached <1 person by the designated year (2030 or 2040). Prob. RS (reactive screening) is calculated as a proportion of the iterations of the dynamic transmission model for which active screening must be re-activated after it has ceased.

is in situ; if AS is stopped after three years of zero detections, there is up to a 62% probability that RS would be necessary (in Kwamouth) in the absence of VC, but at most 19% of VC simulations result in RS (in Budjala).

**Health outcomes**. The outputs of the transmission model were inputs in a probability tree model of disease outcomes (see "Methods" and Fig. 2), from which cases, deaths and disability-adjusted life-years were calculated by standard conventions(for further elaboration on DALYs, see Supplementary Methods, section A.3)[27]. The health impact and net costs of each strategy between 2020 and 2040 are shown in Table 3.

Yasa Bonga, Boma Bungu, and Budjala are each predicted to have an average of ≤5 cases and gHAT-related deaths over the next 20 years. Mosango is predicted to have more cases (≤23) and deaths (≤13) in the absence of VC. Kwamouth has the most predicted cases and deaths, although the burden may be cut by three-quarters with the deployment of VC. In terms of DALYs, Kwamouth sustains the worst burden even under the most ambitious strategy (1718 DALYs, 95% PI: 562–3656) compared to the least ambitious strategy in moderate-risk Mosango (426 DALYs, 95% PI: 32–1418), and any strategy in low-risk Boma Bungu (11–17 DALYs). Intermediate outcomes of treatment are found in Supplementary Methods, section A.2.3: 97% of treated stage-1 patients and 93% of treated stage-2 patients will be cured; an additional 1% and 3%, respectively, will be cured after sustaining severe adverse events; between 2% and 4% of patients will need rescue treatment; and fewer than 1% of all treated cases will die. Further details of the outcomes under different time horizons are found in Supplementary Tables 16, 17 and on the project website: https://hatmepp.warwick.ac.uk/5HZCEA/v2/.

**Costs**. We devised cost functions to calculate costs from the perspective of the healthcare system as a whole (including all levels of government and donors) in 2018 US$. The total mean costs in each location range between $490,000 (Boma Bungu) and

$5.43 million (Kwamouth), and on a yearly per-capita basis, costs range from $0.20 (Budjala) to $1.97 (Kwamouth) (Table 3). The expected costs of strategy components are shown in Fig. 3; these calculations are derived using the mean costs and the probability that activities have not ceased by a particular year. For all locations except Kwamouth, additional costs in the latter half of the 2030s arise from PS, as most simulations indicate that AS and VC would have ceased by this period. In Yasa Bonga, Mosango, and Kwamouth, about three-quarters of the cumulative costs are expected to be spent within the first five years of the 2020s and then stabilize when AS and VC cease (Supplementary Fig. 6). Costs with the least ambitious (default) strategy in Kwamouth and Mosango are expected to overtake the costs of a strategy with VC by 2040, indicating that such a strategy would be economically neutral or even cost-saving in 20 years. In Boma Bungu and Budjala, where scant screening is undertaken and VC would cease quickly, costs are slow to accumulate throughout the period of study. The cost breakdowns across different investment horizons are shown in Supplementary Fig. 5 and show similar patterns to those in Fig. 3. Interested readers may explore the costs and their breakdowns on the project website: https://hatmepp.warwick.ac.uk/5HZCEA/v2/.

**Cost-effectiveness**. The cost-effectiveness of each strategy in each health zone is displayed in Table 4 and select features are illustrated in Fig. 4, and more traditional cost-effectiveness acceptability frontiers (CEAFs) are shown in Supplementary Figs. 7, 8. The standard measure of cost-effectiveness (or comparative efficiency) is the incremental cost-effectiveness ratio (ICER), defined as the additional cost to avert an additional DALY of disease compared to the next-best strategy. We considered the health impacts and costs in a relatively long-term horizon (2020–2040), discounting at a yearly rate of 3% in accordance with standard conventions[27]. Therefore, strategies are considered "minimum cost" for the least costly strategy over a 20-year horizon in 2018 US$ (costs discounted at 3% annually).

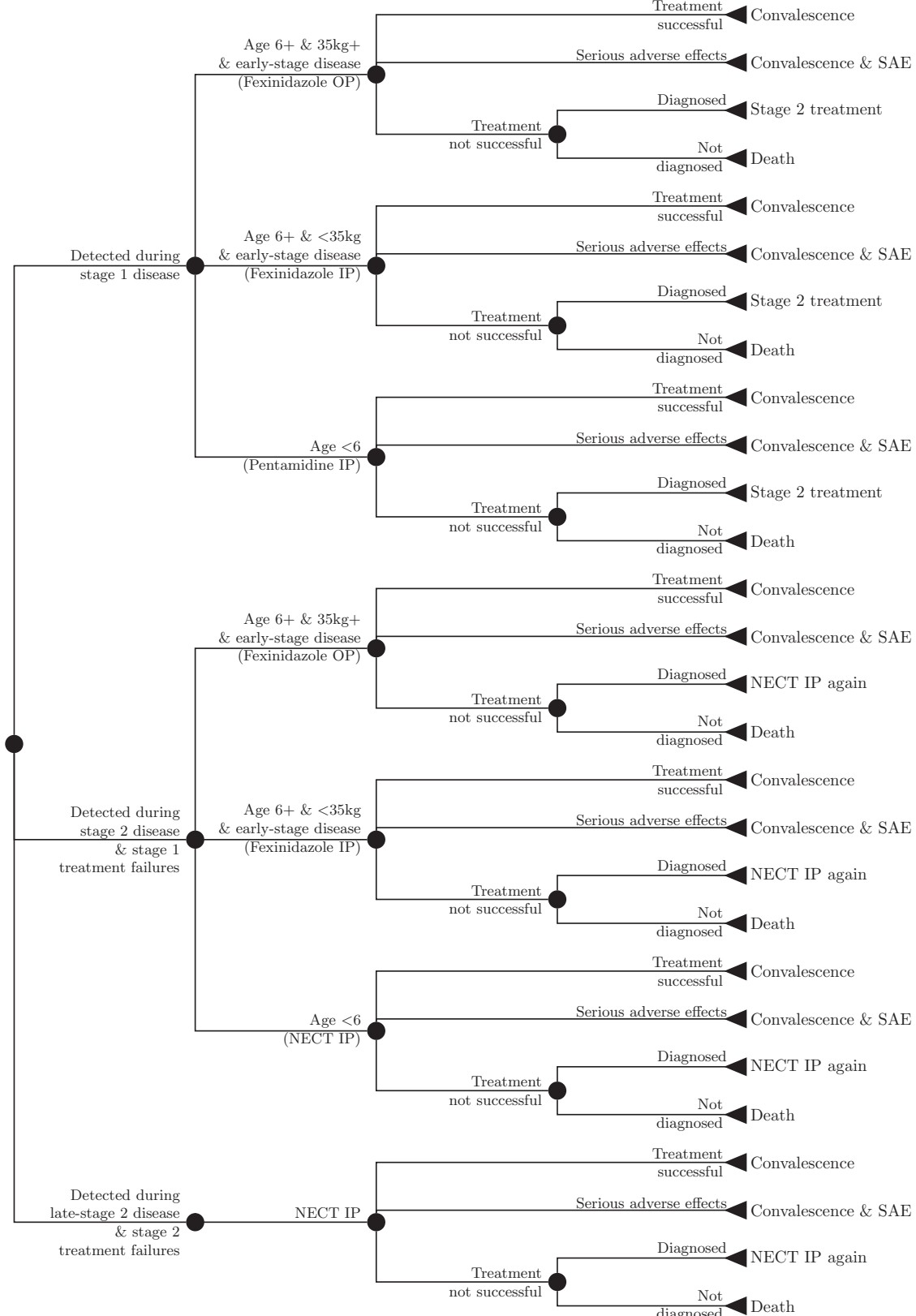

**Fig. 2 Treatment for diagnosed gHAT patients is modeled as a branching tree process of possible health outcomes including eligibility for novel treatment fexinidazole.** Stage 2 disease treatment sometimes applies to stage 1 treatment failures, and some late-stage disease includes some patients who were stage 2 treatment failures. The outcomes of the small proportion of cases that experience unsuccessful treatment are determined by calculating the product of the probability of unsuccessful treatment and the outcome of disease at a later stage of disease. Cases are assumed to go through treatment at most twice. Abbreviations: SAE serious adverse events, IP inpatient care, OP outpatient care, NECT nifurtimox-eflornithine combination therapy.

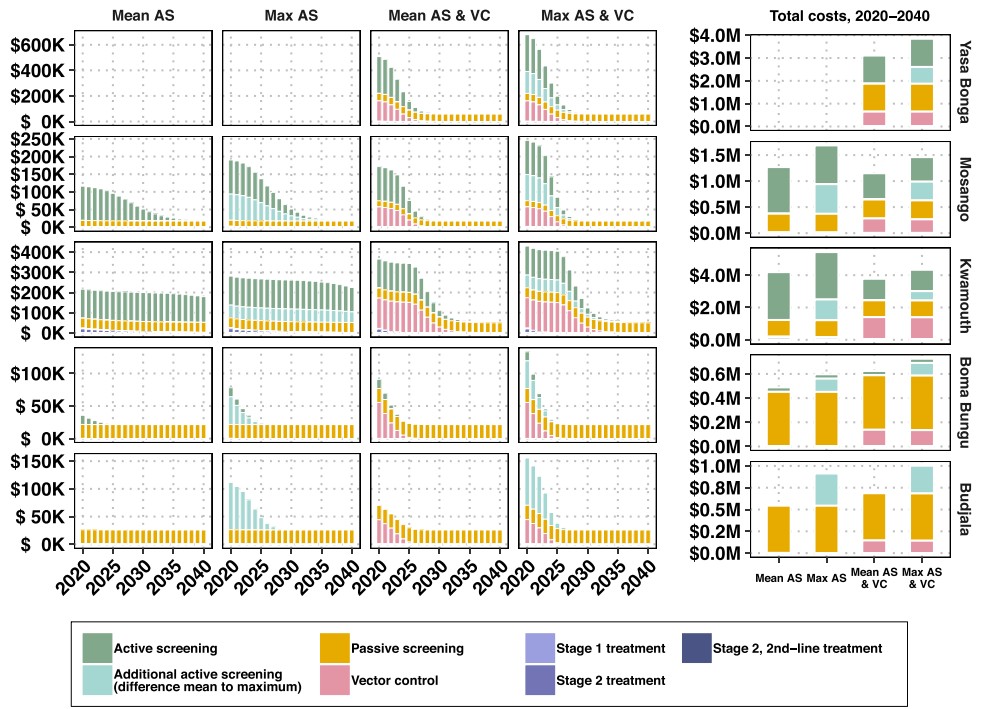

**Fig. 3 Components of annual and cumulative costs, by strategy and location.** Expected costs are the product of the average cost of each component of prevention, detection, and treatment and the probability that activity have not ceased. Displayed costs are not discounted. Treatment costs, indicated in purple, are shown here although they are so small as to be hardly visible.

### Table 3 Summary of effects and costs 2020-2040.

|  | Cases detected (95% PI) | Deaths (95% PI) | DALYs (95% PI) | Total costs ($ millions) (95% PI) | Yearly costs ($) per capita (95% PI) |
|---|---|---|---|---|---|
| Yasa Bonga |  |  |  |  |  |
| Mean AS & VC | 5 (0, 23) | 2 (0, 7) | 62 (1, 240) | 3.11 (1.63, 5.27) | 0.67 (0.35, 1.13) |
| Max AS & VC | 4 (0, 23) | 2 (0, 7) | 62 (1, 242) | 3.84 (1.83, 6.80) | 0.82 (0.39, 1.46) |
| Mosango |  |  |  |  |  |
| Mean AS | 23 (1, 79) | 12 (1, 42) | 426 (32, 1418) | 1.27 (0.62, 2.33) | 0.48 (0.23, 0.89) |
| Max AS | 22 (0, 92) | 8 (0, 28) | 282 (2, 987) | 1.69 (0.75, 3.25) | 0.64 (0.29, 1.24) |
| Mean AS & VC | 9 (0, 41) | 5 (0, 15) | 169 (2, 510) | 1.15 (0.63, 1.85) | 0.44 (0.24, 0.70) |
| Max AS & VC | 10 (0, 54) | 4 (0, 12) | 131 (1, 421) | 1.46 (0.74, 2.46) | 0.56 (0.28, 0.94) |
| Kwamouth |  |  |  |  |  |
| Mean AS | 477 (144, 1,081) | 207 (41, 614) | 7229 (1496, 21,131) | 4.19 (2.88, 6.42) | 1.52 (1.05, 2.33) |
| Max AS | 463 (136, 1,047) | 174 (36, 499) | 6067 (1304, 17,296) | 5.43 (3.64, 8.54) | 1.97 (1.32, 3.10) |
| Mean AS & VC | 116 (41, 235) | 54 (18, 116) | 1890 (628, 4025) | 3.78 (2.49, 5.92) | 1.37 (0.91, 2.15) |
| Max AS & VC | 120 (38, 270) | 49 (16, 105) | 1718 (562, 3656) | 4.33 (2.77, 7.03) | 1.57 (1.01, 2.55) |
| Boma Bungu |  |  |  |  |  |
| Mean AS | 1 (0, 10) | 0 (0, 4) | 17 (0, 149) | 0.49 (0.32, 0.71) | 0.27 (0.18, 0.39) |
| Max AS | 1 (0, 10) | 0 (0, 3) | 13 (0, 109) | 0.60 (0.37, 0.92) | 0.33 (0.21, 0.51) |
| Mean AS & VC | 1 (0, 7) | 0 (0, 3) | 13 (0, 107) | 0.62 (0.39, 0.95) | 0.35 (0.21, 0.53) |
| Max AS & VC | 1 (0, 8) | 0 (0, 3) | 11 (0, 97) | 0.73 (0.43, 1.16) | 0.40 (0.24, 0.64) |
| Budjala |  |  |  |  |  |
| Mean AS | 4 (0, 22) | 5 (0, 18) | 163 (0, 601) | 0.55 (0.36, 0.80) | 0.20 (0.13, 0.29) |
| Max AS | 4 (0, 24) | 2 (0, 8) | 80 (0, 277) | 0.92 (0.45, 1.55) | 0.33 (0.16, 0.55) |
| Mean AS & VC | 2 (0, 12) | 2 (0, 8) | 83 (0, 274) | 0.69 (0.41, 1.06) | 0.25 (0.15, 0.38) |
| Max AS & VC | 3 (0, 19) | 2 (0, 6) | 56 (0, 208) | 1.01 (0.46, 1.68) | 0.36 (0.17, 0.60) |

Two differences should be noted between these estimates and those used for decision analysis shown in Table 4. First, these estimates are not discounted. Second, due to asymmetric distributions, a naive difference in mean costs would not equal the mean differences in costs across simulations—the metric we used in decision analysis. Undetected cases are reflected in deaths, as very few deaths (<1 percent) originate from treated cases. Estimates shown are means and 95% prediction intervals (PI) of the cases, deaths, disability-adjusted life-years (DALYs), and costs across iterations of the dynamic transmission model.

In all health zones but Mosango, the minimum-cost strategy is the current practice; in Mosango, the current practice is "dominated" because the addition of VC yields cost-savings on a 20-year horizon while averting more DALYs than strategies without VC (Mean AS and Max AS). All strategies designated as "dominated" are more costly and less effective at averting DALYs than another strategy.

A second form of dominance is evident in Budjala, where Max AS is designated "weakly dominated". A strategy is "weakly

**Table 4 Summary of cost-effectiveness, assuming a time horizon of 2020–2040.**

| | Cost-effectiveness analysis without uncertainty | | | Net benefit (uncertainty) analysis: Prob. that a strategy is optimal, (conditional on willingness-to-pay) | | | | |
| --- | --- | --- | --- | --- | --- | --- | --- | --- |
| | Cost difference vs comparator | DALYs averted vs comparator | ICER | $0 per DALY averted | $250 per DALY averted | $500 per DALY averted | $1,000 per DALY averted | Prob. EOT by 2030 |
| **Yasa Bonga** | | | | | | | | |
| Mean AS & VC | 0 | 0 | Min Cost | 0.78* | 0.78* | 0.78* | 0.78* | >0.99 |
| Max AS & VC | 671,462 | 0 | 2209,891 | 0.22 | 0.22 | 0.22 | 0.22 | >0.99 |
| **Mosango** | | | | | | | | |
| Mean AS | 0 | 0 | Dominated | 0.38 | 0.33 | 0.29 | 0.24 | 0.79 |
| Max AS | 377,463 | 80 | Dominated | 0.04 | 0.04 | 0.04 | 0.05 | 0.92 |
| Mean AS & VC | −48,090 | 142 | Min Cost | 0.49* | 0.53* | 0.56* | 0.59* | >0.99 |
| Max AS & VC | 237,522 | 165 | 12,215 | 0.08 | 0.09 | 0.1 | 0.13 | >0.99 |
| **Kwamouth** | | | | | | | | |
| Mean AS | 0 | 0 | Min Cost | 0.44* | 0.21 | 0.14 | 0.07 | <0.01 |
| Max AS | 921,216 | 602 | Dominated | 0 | 0 | 0 | 0 | <0.01 |
| Mean AS & VC | 11,632 | 2753 | 4 | 0.49 | 0.65* | 0.68* | 0.69* | >0.99 |
| Max AS & VC | 489,117 | 2861 | 4421 | 0.07 | 0.14 | 0.18 | 0.24 | >0.99 |
| **Boma Bungu** | | | | | | | | |
| Mean AS | 0 | 0 | Min Cost | 1* | 0.99* | 0.99* | 0.99* | >0.99 |
| Max AS | 101,606 | 3 | 40,288 | 0 | 0 | 0 | 0.01 | >0.99 |
| Mean AS & VC | 127,894 | 2 | Dominated | 0 | 0 | 0 | 0.01 | >0.99 |
| Max AS & VC | 223,232 | 4 | 93,060 | 0 | 0 | 0 | 0 | >0.99 |
| **Budjala** | | | | | | | | |
| Mean AS | 0 | 0 | Min Cost | 0.91* | 0.87* | 0.85* | 0.76* | 0.97 |
| Max AS | 335,786 | 47 | Weakly Dominated | 0.04 | 0.03 | 0.03 | 0.03 | >0.99 |
| Mean AS & VC | 131,747 | 45 | 2922 | 0.04 | 0.07 | 0.1 | 0.18 | >0.99 |
| Max AS & VC | 423,280 | 62 | 17,515 | 0.01 | 0.02 | 0.02 | 0.03 | >0.99 |

Cost differences and differences in disability-adjusted life-years (DALYs) averted are relative to the comparator-first strategy listed for each location. Mean DALYs averted and mean cost differences are shown; these estimates are discounted at 3 percent per year in accordance with guidelines. The uncertainty analysis (columns 5–8) shows the probability that a strategy is cost-effective. Strategies marked by an asterisk are optimal strategies at the willingness-to-pay indicated by the column title: the strategies for which the mean net monetary benefit (NMB) is highest, equivalent to the information found in cost-effectiveness acceptability frontiers (CEAFs), which are shown Supplementary Figs. 7, 8. ICER: incremental cost-effectiveness ratio. For an extended discussion of these terms, see the Supplementary Discussion. For a full explanation of the concept of strong and weak dominance, see the Supplementary Note 1: Glossary of Technical Terms.

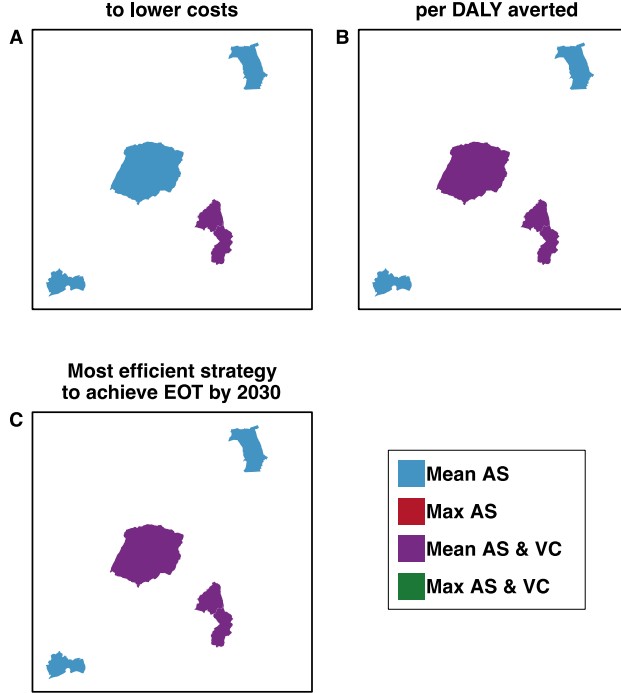

**Fig. 4 Maps of preferred strategies according to economic or budgetary goals for 2020–2040.** Maps (**A**) & (**B**) show the optimal strategies depending on willingness-to-pay (WTP). The text indicates the probability that the optimal strategy will lead to elimination of transmission (EOT) by 2030. Map (**C**) shows the optimal strategy that has at >90% probability of EOT by 2030 and shows the incremental cost-effectiveness ratio (ICER) of the indicated strategy (Mean AS for all locations except Yasa Bonga, where it is Mean AS & VC). Maps are not drawn to scale. Maps with time horizons 2020–2030 and 2020–2050 are in the Supplementary Figs. 9 and 10.

dominated" if the next more expensive strategy (in absolute terms) has a lower ICER—indicating that although more expensive, the latter is more efficient on a per-DALY-averted basis than the former. In Budjala, the Max AS strategy is weakly dominated by the Max AS & VC strategy, because although the latter is more expensive the additional cost is better justified on a per-DALY basis. For an extended discussion on dominated and weakly dominated strategies, see the Supplementary Discussion. In short, both dominated and weakly dominated strategies present inefficiencies.

A strategy is considered cost-effective, though not cost-saving, when it maximizes DALYs averted while having an ICER below an acceptable threshold, considered as the willingness-to-pay (WTP) threshold. The WTP value is denominated in costs-per-DALY averted and we do not prescribe any specific WTP value, as per WHO recommendations. We leave the choice of threshold for the policy-maker, though we note that other analyses have shown that the WTP in DRC for any new strategy is approximately $5–230 per DALY averted[28], and past guidelines have delineated cost-effectiveness threshold as equal to the GDP per capita of the country[27], which was $557 in DRC in 2018.

In all places but Kwamouth, the ICER for strategies other than the minimum-cost-strategy are moderately high, indicating low comparative efficiency for settings in DRC. In Kwamouth, the Max AS & VC strategy maximizes DALYs averted, at an ICER of $4421, which is considered too high by any current or historical metric, but the Mean AS & VC strategy has an ICER of $4, below historical WTP thresholds, so it satisfies both criteria of maximizing DALYs averted and staying under the WTP threshold.

**Cost-effectiveness and elimination of transmission by 2030.** In Yasa Bonga and in Boma Bungu, EOT is estimated to occur by 2030 with >99% probability, and any more ambitious operations would be to detect existing cases—yielding high ICERs. In Budjala, perhaps because of minimal AS coverage, there is a 36% probability that RS would be deployed in Budjala after cessation (see Table 2), and while VC would nearly halve that prospect, it would come at a steep cost of $2,932 per DALY averted. Last, while in Mosango, adding VC to the current practice of AS is predicted to both save money and increase the probability of EOT by 2030 from 79% to >99%, in Kwamouth the same shift makes EOT by 2030 possible, but at a cost of $4 per DALY averted.

**Cost-effectiveness in the presence of probabilistic uncertainty.** While the ICER provides an intuitive measure of efficiency, it does not integrate uncertainty in decision-making, so we present the ICERs here for comparability with other literature. We base our conclusions of cost-effectiveness on the results within the net benefits framework, which examines the efficiency of gHAT end-game strategies in the presence of uncertainty (see Supplementary Methods, section A.5, and Supplementary Note 1: Glossary of Technical Terms). We consider a strategy "optimal" or cost-effective if the net monetary benefit (NMB) is maximum at a given willingness-to-pay value.

In Yasa Bonga, Boma Bungu and Budjala, the predicted minimum-cost strategy has a 97–99% probability of EOT by 2030. After accounting for uncertainty, the current practice (Mean AS & VC) in Yasa Bonga is the optimal strategy across all WTP thresholds <$1000 in >78% of simulations. In Budjala, there is even more support (76–91% of simulations) indicating that the least ambitious intervention (Mean AS) is optimal across all WTP values. In Mosango, Mean AS & VC is cost-saving with a 49% probability. In Kwamouth, >65% of simulations favored the addition of VC to Mean AS for WTP values above $250, and in fact, was cost-saving in 42% of simulations. Cost-effectiveness acceptability curves, expressing the same information as in Table 4 but in a more conventional format within the net benefits literature, are shown in Supplementary Fig. 7.

**Scenario analyses.** In addition to the uncertainty due to probabilistic uncertainty, we perform scenario analyses to understand the robustness of our findings when the uncertainty in assumptions cannot be quantified probabilistically.

In Mosango and Kwamouth, VC would both raise the probability of EOT by 2030 and minimize the probability of RS. However, these conclusions are contingent on assumptions around VC operations. In Mosango, the potential for cost-savings is contingent on the assumption that an operation needs to be deployed across 100 km of riverbank only. In fact, with a small operation, VC remains cost-saving even if tsetse population reduction is as low as 60% in one year (see Supplementary Fig. 11). However, if an operation is needed in Mosango that is closer in extent to the one in Yasa Bonga (across 210 km of riverbank dotted with 40 targets per kilometer of riverbank) then VC would be optimal if WTP would exceed $1488–2051 per DALY averted, depending on the effectiveness of VC to reduce tsetse populations.

In Kwamouth, we found that if target density must be doubled, WTP must exceed $236–343 per DALY averted after considering deviations in VC effectiveness (see Supplementary Fig. 12). Last, we performed additional scenario analyses to examine the impact of our default assumptions around time horizons, discounting (Supplementary Methods, section A.6). The results across shorter (2020–2030) and longer (2020–2050) time horizons are very similar (see Supplementary Figs. 9, 10 and Supplementary

Tables 23, 24). The results assuming no discounting of costs or effects (Supplementary Fig. 8) lead to similar recommendations for WTP values <$1000.

Interested readers can explore the results of all scenario analysis on the project website: https://hatmepp.warwick.ac.uk/5HZCEA/v2/.

## Discussion

Applying a decision-analytic framework across five health zones of DRC we have assessed current efforts to control gHAT, as well as the efficiency of alternative strategies to transition beyond suppression to EOT. The transmission model predicted substantial declines in observed gHAT cases and the underlying transmission in all locations using any strategy, but the cumulative burden of disease and the capacity to reach EOT by 2030 varied considerably.

In Mosango and Kwamouth the addition of VC to Mean AS is predicted to expedite EOT and be cost-effective for low WTP; while the addition of VC would ensure EOT while saving costs in Mosango, the WTP to choose the strategy that ensures EOT is only $4/DALY (Table 4). However, in Boma Bungu and Budjala current practice appears sufficient to ensure EOT by 2030 despite Boma Bungu's "moderate" incidence and Budjala's low AS coverage (<0.5%) (Tables 1 and 4). In Yasa Bonga, where VC has been coupled with a relatively high coverage rate (57% of the 2016 population) EOT is estimated to have occurred already (Table 2).

Although the expected total costs in each location vary between $480,000 in Boma Bungu to $5.26 million in Kwamouth, the optimal strategies have a expected cost of at most $3.71 million (Table 3). Per person, the optimal strategies would not exceed $1.37 per year per person protected (Table 3)—comparable to many other global health interventions[15]. Notably, while healthcare costs and DALYs can be considered narrow criteria to justify investments in elimination, we chose to conduct an analysis that would allow us to understand the relative efficiency of these strategies by the same rubric employed to assess the cost-effectiveness of other disease programs; we found that local gHAT elimination is possible and cost-effective at modest economic costs.

Additionally, multiple CEAs across health zones could inform allocative efficiency in case of budget constraints, as it shows that, among these five health zones, the most efficient use of scale-up dollars would first be in Mosango and then in Kwamouth, as the addition of VC would be cost-saving in Mosango while averting disease and it would be cost-effective at $4/DALY averted in Kwamouth.

While a full analysis of past costs is nearly impossible due to the heterogenous nature of funding for gHAT activities, and because national health accounts contain no details on a disease like gHAT, our cost parameters were informed by recent cost studies in Yasa Bonga and Mosango, therefore updating the costs used in the past[16,18,19,29]. A decomposition of the costs showed that while there are contributions from varied factors, some broad patterns emerged. First, AS costs and, when applicable, VC costs will play a large role in overall costs everywhere except in Budjala, where no more than PS and minimal AS is recommended. Second, cessation of AS and VC activities means costs are expected to decrease during the 2020's, and in Kwamouth, investments in VC in the early 2020's could be recovered by the late-2030's (Supplementary Fig. 6). Therefore, while cost-effectiveness analysis cannot take into account all potential elements of implementation, such as the political will to implement certain components of the strategies proposed, analysis is insightful to understand the comparative efficiency within any two strategies or to raise funds, as we have presented the time-scale for a return on investment.

Importantly, we have found that our conclusions are quite robust to the choice of time-horizon as well as discount rate, and our framework has accounted for parameter uncertainty. We have chosen to consider a 20-year horizon with supplemental results for a 10- and 30-year horizon because strategies with long-term consequences like EOT would be under-appreciated in a short horizon, but conclusions do not differ substantially (see Supplementary Tables 23, 24 and Supplementary Figs. 7, 8).

Simulation results presented here suggest that local EOT of gHAT is epidemiologically and operationally feasible across different risk settings with the current toolbox. As expected, higher-burden health zones expect later EOT, but VC raises the possibility and substantially expedites EOT, as seen across DRC[21]. The model utilized here did not explicitly include potential hindrances such as asymptomatic human infection or animal reservoirs[30], however, the long-tailed distribution used for disease progression captures possible long-term asymptomatic carriage. Skin-only infections or asymptomatic individuals whose infection may self-resolve would require further model modifications and ongoing modeling work will examine potential risks and impact. Previous modeling analyses indicate that the existence of animal reservoirs may not greatly alter timelines to EOT, especially if VC is implemented[6,23].

This is the first analysis where fexinidazole is the default treatment, but to reflect the caution in the WHO guidelines, 65% of simulated patients were assumed to be treated with NECT or on an in-patient basis due to late-stage detection, low body weight, or the young age of the patient (see Supplementary Table 2 and Supplementary Note 4: Parameter Glossary, sections G.6.1–G.6.3). The prospect of a single-dose oral treatment for gHAT (acoziborole), currently in trials, indicated for broad swaths of the population could further bolster the impact of medical interventions if it allows the treatment of gHAT suspects without confirmation; evaluating the impact of a drug that overcomes the limitations of the current treatment arsenal is beyond the scope of this analysis, and a careful treatment of the matter ought to be pursued[13]. Similarly, the impact of more specific diagnostics that may become available could change the conclusions of this cost-effectiveness analyses, as it is comparative efficiency that it informs.

Previous cost-effectiveness analyses have underlined key factors associated with suppression and elimination of gHAT, but this is the first study to confront the question of cost-effectiveness with data from five specific locations[16,31,32]. An analysis by Bessel et al. shows that RDTs in DRC are cost-effective when employed within AS and PS[31], and an analysis by Davis et al. shows that once-per-year AS campaigns within endemic villages are cost-effective to control disease[32]. Only one other analysis, performed by Sutherland et al., examined the combination of multi-faceted strategies, and they found that elimination strategies are likely cost-effective at a WTP of $400–1500 (in 2013 US$) in high- and moderate-incidence places[16]. One important difference to the analysis by Sutherland and colleagues is that we assumed yearly (rather than biennial) AS campaigns at coverage levels that match historical averages (rather than a fixed value of 80% of the population—a high value relative to historical coverage observed in DRC)[16]. We also assumed that AS would persist for three years after the last detected case in a health zone, irrespective of the initial incidence of the health zone, whereas Sutherland et al. assumed no AS in low-incidence health zones and therefore had to recommend VC activities to reach EOT in those contexts. In our lower incidence health zones (Boma Bungu and Budjala), the presence of AS meant that EOT could be reached without VC (Table 4).

Our analyses reinforce previous findings that VC would be both an expedited method of achieving EOT and cost-effective in

one moderate-incidence health zone (Mosango) and one high-incidence health zone (Kwamouth). However, determining the amount of VC necessary to reach a desired tsetse population reduction is a complicated task, as it depends on local ecology. For transparency, we have shown cost-effectiveness results in a three-way sensitivity analysis in places where VC might be warranted and was not previously in place (Mosango and Kwamouth; Supplementary Tables 25, 26 and Supplementary Figs. 11, 12). In Mosango, a modest VC operation could be cost-saving, but these results are contingent on the geographic distribution of current cases as well as on the sensitivity of the tsetse population to targets (Supplementary Fig. 11). Because Kwamouth is substantially larger than other health zones, the geographic clustering of cases will be important to determine whether all high-risk areas can be addressed in a cost-effective way (Supplementary Fig. 12).

Our findings took into account historical improvements in PS, which was made possible by more recent data and novel model calibration; this element of the strategies has difficult-to-quantify impacts that might explain some of the difference between our results and previous findings. The fact that our analysis was more optimistic about current practice in lower-incidence health zones underscores the potential of a well-equipped health system that can serve self-presenting gHAT cases. Future analyses are warranted on the impact of gHAT screening integrated into the primary care system.

Within the last year, DRC has contended with the emergence of COVID-19, which triggered an untimely but temporary disruption to AS activities[33]. But while these interruptions are lamentable, the EOT goal could still be within reach as long as disruptions remain short[34]. While reported gHAT cases now number under 1000 globally, there remain 51 million people in environments that could sustain gHAT transmission[11]. The epidemic potential of gHAT in locations as resource-constrained as DRC underscores the importance of careful deliberation around strategies of gHAT elimination.

## Methods

**Transmission model.** We employed a previously published dynamic, transmission "susceptible-exposed-infected-recovered-susceptible (SEIRS)" model, combined with a probability tree of treatment (Fig. 2), in a manner fully illustrated in Supplementary Fig. 4. While the model and equations are detailed elsewhere[17,21], briefly, we simulated gHAT infection and vector transmission in a deterministic compartmental model using a set of ordinary differential equations which represent average expected dynamics. Stochastic features of the system—corresponding to probabilistic, or chance events were included for case detections and deaths and were simulated by sampling from the prevalence of infected individuals for reporting to care in fixed facilities or mobile teams. Specifically, new infections and person-years spent in stage 1 and 2 are deterministic outputs, dependent on the parameter inputs, but case reporting and deaths also include random sampling using the beta-binomial distribution. The absolute population that is seen by active screening is fixed, although the population is projected to increase by 3% per year. The key outputs of the dynamic and diagnostic models include mortality in undetected cases, detected cases in stages 1 and 2, and DALYs before and after presenting to care for all interventions. Further details on the development of the model are found in Supplementary Methods, section A.2.1.

**Strategies.** All strategies (Fig. 1) are constituted of a combination of interventions, detailed below:

- Active screening (AS): at-risk village populations are screened each year by mobile teams and suspected cases are confirmed before treatment. In two strategies (1 and 3, "Mean AS") the number of people screened annually per health zone is equal to the mean number of individuals screened during 2014–2018 (the last five years for which there are data). In two other strategies (2 and 4, "Max AS"), the number of people screened annually is the maximum coverage achieved in that health zone in any single year between 2000 and 2018.
- Passive screening (PS): testing of symptomatic patients who self-present to local health centers with gHAT diagnostics, followed by confirmation and treatment[3].
- Large-scale vector control (VC): twice-yearly deployment of tiny targets to control the population of tsetse, with an assumed 80% reduction after one

year, consistent with the lower bound efficacy estimates of field studies[5,6,8,9]. In Yasa Bonga, VC reduced the tsetse population by 90% in one year, so our assumption in that health zone is a bit higher. This intervention is assumed to span the areas with ongoing transmission in the previous five years, in keeping with existing operations.

In all strategies, AS (at either "Mean" or "Max" coverage) and PS are in place, with supplemental VC deployed in Strategies 3 and 4 (see Fig. 1). In Yasa Bonga, VC has taken place since mid-2015, so only the two VC strategies (3 and 4) were considered.

The transition between the suppression and post-elimination phases, not included in the previous projections[21], is simulated as the cessation of both AS and VC after three consecutive years of zero detected cases by any screening modality (AS or PS). Should a new case present to fixed health facilities (through PS), reactive screening (RS) begins the following year: equivalent in form and intensity to the previous AS, RS is simulated until there are two consecutive years of zero case detection by any mechanism (either AS or PS). The availability of screening in health facilities (PS) is assumed to remain constant for the duration of our simulations, even after cessation of AS and VC and EOT.

**Feasibility of EOT by 2030.** For each realization we record the year that EOT is achieved using a proxy threshold of less than one new infection per year in the health zone. We presented feasibility with a variety of metrics indicating the timing of EOT as well as the uncertainty around the goal by 2030:

- The mean and 95% prediction interval for the year of EOT is calculated by taking the year in which cases fall under 1 in each iteration of the model and calculating the arithmetic mean and 2.5th and 97.5th percentile of the sample.
- The probability of EOT by 2030 was the percentage of iterations in which the year of EOT was 2029 or earlier.
- The estimates for year of AS cessation—and if applicable, VC cessation—were calculated by taking the year when AS first stopped occurring, and calculating the mean and 95% prediction intervals as above.
- The probability of RS by 2050 was the percentage of iterations in which there was at least one year AS after AS had first ceased (after at least one year of no AS).

**Cases, deaths, and treatment outcome model.** Using a probability tree (Fig. 2), we simulated the treatment process separately for stage 1 and stage 2 disease, sorting patients into the WHO-recommended treatment, then into treatment success or failure, correct diagnosis in the event of treatment failure, and progression to rescue treatment or to death (for further details, see Supplementary Methods, section A.2.3)[4]. WHO-recommended treatment consists of fexinidazole on an inpatient or outpatient basis depending on patient characteristics, and pentamidine or nifurtimox-eflonithine combination therapy (NECT) when fexinidazole is contraindicated (see Supplementary Methods, section A.2.3 for a breakdown of treatment groups)[4].

Disability-adjusted life years (DALYs) were calculated for both detected and undetected cases, which were assumed to all be fatal (Supplementary Methods, section A.3). DALYs are the sum of the years of life lost to disability (illness) (YLD), weighted by the severity of the disease, and the years of life lost by fatal cases (YLL)[27,35].

Uncertainty arose from two sources. First, the uncertainty in dynamic model parameters fit to historical data, formally drawn by an adaptive MCMC algorithm (for more information, see[17]). Second, uncertainty surrounding treatment outcomes in literature-based estimates, which was formalized by assigning probability distributions according to standard conventions (for details, see Supplementary Note 4: Parameter Glossary, section G.1). Uncertainty in final estimates of cases, deaths, and treatment outcomes were therefore presented as means and 95% prediction intervals, calculated by taking the 2.5th and 97.5th percentiles of the simulated samples.

**Costs.** Based on field experience from Yasa Bonga and Mosango[18,19], we devised a cost function to calculate the costs of detection via AS and PS, prevention via VC, and treatment (see Supplementary Methods, section A.4, and parameters in Table 5 and Supplementary Note 4: Parameter Glossary). Screening costs include diagnostics, mobile teams, and fixed facilities (more details can be found in Supplementary Methods, section A.4). Costs of screening were a mix of fixed and variable costs: fixed costs calculated according to health facilities or the mobile teams needed, staff training, and variable costs according to the people screened by health facility or mobile team, and people confirmed.

Treatment costs include staging via lumbar puncture (when indicated), and drug administration; these costs were based on the literature. VC costs were scaled according to the geographic expanse treated (in terms of km of riverbank) and the density of targets placed along riverbanks. Intermediate and subtotal costs of each activity are described in Supplementary Methods, section A.4. Because no published estimates of VC cost exist from DRC, we used published estimates from Uganda. A more recent study published for Chad activities showed that operations were of similar cost[36].

**Table 5 Model Parameters. For further details and sources, see Section G.**

| Variable Description | Statistical Distribution | Descriptive Summary Mean (95% CIs) |
|---|---|---|
| Screening parameters | | |
| Population | Fixed value | HZ-specific (see Table 1) |
| PS: coverage of the population per facility | Beta (14, 2094) | 0.007 (0.004, 0.010) |
| PS: number of facilities | Fixed value | HZ-specific (see Table 1) |
| AS: coverage | Fixed value | HZ-specific (see Table 1) |
| AS: coverage, enhanced | Fixed value | HZ-specific (see Table 1) |
| AS: coverage by each team per year | Normal (60000, 10000) | 60,055 (40,448, 79,471) |
| CATT algorithm: diagnostic specificity | Beta (31, 2) | 0.94 (0.84, 0.99) |
| RDT algorithm: diagnostic sensitivity | Beta (230, 1) | 1.00 (0.98, 1.00) |
| RDT algorithm: diagnostic specificity | Beta (226, 31) | 0.88 (0.84, 0.92) |
| CATT algorithm: wastage during AS | Beta (8, 92) | 0.08 (0.03, 0.14) |
| RDT algorithm: wastage during PS | Beta (1, 99) | 0.01 (<0.01, 0.04) |
| Screening cost parameters | | |
| AS: capital costs of a team | Gamma (81.02, 114.54) | 9276 (7378, 11,375) |
| AS: fixed management costs of a team | Gamma (63.31, 630.94) | 39,955 (30,845, 50,435) |
| CATT algorithm: cost per test used | Gamma (25.19, 0.02) | 0.52 (0.34, 0.75) |
| Staging: lumbar puncture & lab exam | Gamma (2.42, 3.66) | 8.90 (1.45, 23.20) |
| Confirmation: microscopy | Gamma (8.47, 1.27) | 10.68 (4.70, 18.84) |
| RDT algorithm: costs per test used | Gamma (8.47, 0.19) | 1.60 (0.71, 2.83) |
| Variable management costs (PNLTHA mark-up) | Uniform (0.1, 0.2) | 0.15 (0.10, 0.20) |
| PS: capital costs of a facility | Gamma (8.47, 209.8) | 1777 (778, 3157) |
| PS: fixed recurrent management costs | Gamma (8.47, 985.55) | 8368 (3743, 14,965) |
| Treatment parameters | | |
| Proportion of cases age < 6 | Beta (152.53, 2427.9) | 0.06 (0.05, 0.07) |
| Proportion of cases weight < 35 kg among age > 6 | Beta (8.3, 359.6) | 0.02 (<0.01, 0.04) |
| Proportion of S2 cases that are severe | Beta (76.93, 44.87) | 0.63 (0.54, 0.72) |
| Age of death from infection | Gamma (148, 0.18) | 26.63 (22.41, 31.08) |
| Length of hospital stay: NECT treatment | Fixed value | 10 |
| Length of hospital stay: fexinidazole treatment | Fixed value | 10 |
| Pr. of relapse: pentamidine | Beta (50.3, 665.48) | 0.07 (0.05, 0.09) |
| Pr. of relapse: NECT | Beta (15.87, 378.55) | 0.05 (0.02, 0.08) |
| Pr. of relapse: fexinidazole | Beta (9.49, 496.54) | 0.02 (<0.01, 0.03) |
| SAE: pentamidine treatment | Beta (1.43, 551.42) | 0.002 (<0.001, 0.008) |
| SAE: NECT treatment | Beta (40.88, 367.8) | 0.05 (0.03, 0.08) |
| SAE: fexinidazole treatment | Beta (3, 261) | 0.01 (<0.01, 0.03) |
| Days lost to disability due to S1 disease | Gamma (21.89, 26.07) | 569.21 (355.50, 831.29) |
| Days lost to disability due to S2 disease | Gamma (22.18, 12.38) | 275.57 (172.46, 401.20) |
| Days lost to disability due to SAE | Gamma (1.22, 2.38) | 2.96 (0.14, 9.99) |
| Treatment cost parameters | | |
| Hospital stay: cost per day | Gamma (5.81, 0.24) | 1.39 (0.50, 2.71) |
| Outpatient consultation: cost | Uniform (1.37, 3.33) | 2.34 (1.42, 3.28) |
| Course of pentamidine: cost | Fixed value | 54 |
| Course of NECT: cost | Fixed value | 460 |
| Course of fexinidazole: cost | Fixed value | 50 |
| Drug delivery mark-up | Beta (45, 55) | 0.45 (0.35, 0.55) |
| Vector control parameters | | |
| Linear km of targets | Fixed value | HZ-specific (see Table 1) |
| Targets per km | Fixed value | HZ-specific (see Table 1) |
| Replacement rate of targets per year | Fixed value | 2 |
| Vector cost parameters | | |
| Operational cost per km | Gamma (8.47, 14.17) | 120.28 (53.33, 212.26) |
| Deployment cost per target | Gamma (8.47, 0.54) | 4.57 (2.02, 8.26) |
| DALY parameters | | |
| Life expectancy | Fixed value | 60.02 |
| Disability weights: S1 disease | Beta (22.96, 147.21) | 0.14 (0.09, 0.19) |
| Disability weights: S2 disease | Beta (18.37, 15.63) | 0.54 (0.37, 0.70) |
| Disability weights: SAE | Uniform (0.04, 0.11) | 0.08 (0.04, 0.11) |

*CIs* confidence intervals, *AS & PS* active and passive screening, respectively, *VC* vector control, *PNLTHA* Programme de Lutte contre la Trypanosomie Humaine, *NECT* nifurtimox-eflornithine combination therapy, *CATT* card agglutination test for trypanosomiasis, *S1 & S2* stage 1 & 2 disease, *HZ* health zone, *DALYs* disability-adjusted life-years, *SAE* severe adverse events.

Costs were from the perspective of the healthcare system collectively (including the government at all levels and donors). Costs were parameterized using values from the literature converted to 2018 US$ values (see Supplementary Note 4: Parameter Glossary, section G.1). Start-up costs were annualized. Uncertainty surrounding costs was formalized by assigning probability distributions to each parameter according to standard conventions and drawing Monte Carlo samples (for details see Supplementary Note 4: Parameter Glossary). Uncertainty in final estimates of costs were therefore presented as means and 95%, calculated by taking the 2.5th and 97.5th percentiles of the samples.

**Cost-effectiveness analysis**. We computed incremental cost-effectiveness ratios (ICERs) to perform our initial cost-effectiveness analysis (illustrated in Supplementary Fig. 4) using the mean DALYs averted and mean costs for each strategy

compared to the current strategy (Mean AS for all but Yasa Bonga, where the current strategy is Mean AS & VC).

To account for parameter uncertainty in the economic evaluation, we adopted the net benefits framework, which expresses the probability that an intervention is optimal at a range of willingness-to-pay thresholds (WTP). We consider a strategy "optimal" or cost-effective if the net monetary benefit (NMB), or the difference between the monetary value of DALYs averted and the additional costs (versus the comparator) is favorable (maximum) in the following formulation:

$$NMB = \Delta DALY \times WTP - \Delta Costs$$

where WTP is the willingness-to-pay, or the cost-effectiveness threshold. Prediction intervals of ICERs are mathematically problematic[37], so convention dictates that we show the probability that a strategy is optimal at range of WTP values, for which we chose $0–1000 in $250 increments. Although WTP values should be designated by the countries, we note that some values for DRC in the literature are approximately $5–230 per DALY averted[28], or $557 (equivalent to the country's GDP per capita in 2018)[27]. For further elaboration on the framework and our implementation, see Supplementary Methods, section A.5.

We also performed scenario analyses to examine the impact of our default assumptions around time horizons, discounting, and on the efficacy and cost of VC (see Section A.6).

Analyses were performed using Matlab 2018b and R version 3.6. Shapefiles used to produce the maps are available under an ODC-ODbL licence at https://data.humdata.org/dataset/drc-health-data. Computational considerations are detailed in the Open Science Framework repository for this analysis at https://osf.io/xbwte. For interested readers and policy-makers, we created a project website to showcase results and sensitivity analysis: https://hatmepp.warwick.ac.uk/5HZCEA/v2/.

**Reporting summary**. Further information on research design is available in the Nature Research Reporting Summary linked to this article.

## Data availability

Information about the WHO HAT Atlas data used for fitting is described in Crump et al.[17] and screening data from the five health zones in this study were used to inform future potential screening coverage and were obtained through the WHO HAT Atlas. Data cannot be shared publicly because they were aggregated from the World Health Organization's HAT Atlas which is under the stewardship of the WHO; our data-sharing agreement does not allow us to share that data. WHO HAT Atlas include identifiable data. Data are available from the WHO (contact neglected.diseases@who.int or visit https://www.who.int/health-topics/human-african-trypanosomiasis/) for researchers who meet the criteria for access to confidential data, including secure computational facilities and an existing relationship to the national sleeping control program of the DRC. Time-frame for response would depend on the WHOs timelines and workloads. Clinical outcomes and costs (listed in Table 5) were simulated using estimates from the literature and are described in Supplementary Note 4: Parameter Glossary. Assumptions and estimates were parameterized according to conventions in the economic evaluation literature[38].

## Code availability

Full access to the code via Open Science Framework: https://osf.io/xbwte/.

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

## Acknowledgements

The authors thank PNLTHA in DRC for original data collection, WHO for data access (in the framework of the WHO HAT Atlas[11]), Cyrus Sinai and Nicole Hoff from UCLA Fielding School of Public Health for providing health zone level shapefiles (current versions can be found at https://data.humdata.org/dataset/drc-health-data). Calculations were performed using the sciCORE (http://scicore.unibas.ch/) Scientific Computing Center at University of Basel. The authors also thank collaborators for their valuable insights on specific topics in this article: Andrew Hope, Paul Bessel, Iñaki Tirados, and Alex Shaw. This work was supported by the Bill and Melinda Gates Foundation (www.gatesfoundation.org) through the Human African Trypanosomiasis Modeling and Economic Predictions for Policy (HAT MEPP) project [OPP1177824] (MA, CH, REC, PB, MJK, KSR, and FT), through the NTD Modeling Consortium [OPP1184344, OPP1156227, and OPP1186851] (KSR and MJK), and through the TRYP-ELIM-BANDUNDU project [OPP1155293] (EMM and RS) and the Directorate-general Development Cooperation and Humanitarian Aid (EMM). The funders of the study had no role in study design, data collection, data analysis, data interpretation, or writing of the report.

## Author contributions

M.A., C.H., R.E.C. and K.S.R. developed the software and performed the analyses. M.A. and P.B. visualized the results. E.M.M. and R.S. curated the data. C.H. and M.A. analysed the data. M.A., R.S. and K.S.R. developed the methods. M.A., K.S.R. and F.T. wrote the original draft. K.S.R., M.J.K., and F.T. conceptualized the study. All authors reviewed and approved the final version for publication.

## Competing interests

The authors declare no competing interests.
