## [Peer Review File · Nature Communications]

Title: Cost-effectiveness of sleeping sickness elimination campaigns in five settings of the Democratic Republic of CongoREVIEWER COMMENTS

Reviewer #2 (Remarks to the Author):

I appreciate very much this opportunity to review this important manuscript. This study evaluates the cost-effectiveness of eliminating Human African trypanosomiasis from the Democratic Republic of Congo. It assesses the population impact and cost-effectiveness of four scenarios that involved passive screening, active screening and vector control and their combinations. Overall, I have found this study to be very robust, both their data source and the analytic approach they employed. The discussion follows closely from their findings. In particular, the key strength their prediction of the probability of achieving EOT by the year 2030. I agree the paper has important relevance to public health and its audience.

I have a few major concerns:

1. The major concern is the method of models; in the main text, it has clearly stated that an SEIRS deterministic model has been used. In the appendix, it appears that it is a decision-analytic model was used, and the SEIRS is embedded in the decision tree model. It is unclear how the two models are working together (Figure S2)? In particular, the SEIRS model uses a set of differential equations. This set of equations was not presented anywhere in the main text or the appendix?
2. Further, the model structure for each scenario is essentially identical. However, it is unclear how the impact of each of these intervention strategies was assessed? For example, how vector control would change the transmission probability or force of infection in the SEIRS model?

Minor comments:

3. I realise the authors used lots of parameters from Uganda since DRC-specific data may not be available (Appendix S79-S85). Could the authors comment on the accuracies of using data from Uganda?
4. It would be interesting to use probability sensitivity analysis to investigate how the change of WTP may impact the probability of cost-effectiveness. Could the author include this information (PMID: 32971056, Figure 2). Further, would a different WTP also impact the probability of achieving EOT by 2030?

Reviewer #3 (Remarks to the Author):

Main comment:

The authors conducted a cost-effectiveness and net-benefit analysis of four strategies for elimination of Gambiense human African trypanosomiasis (gHAT) in the Democratic Republic of Congo. A strength of

their analysis is that it's based on impressive data on the epidemiology of gHAT and health program performance in the five settings in recent years. Their data on actual health programs also raises a concern about perpetuating inequity among the sites.

The results favor current practice (passive screening and mean active screening) in 2 of 4 settings where vector control is not currently implemented. In those 2 settings, Boma Bungu and Budjala, the percentage of the population covered by active screening is relatively low (7.2% and 0.41% respectively), and incidence is low (1.37 and 0.05 per 10,000 population respectively). The main differences between Mosango, where adding vector control is cost effective, and Boma Bungu, are that the percentage of the population with active screening in 2016 was 34% in Mosango (i.e. 5 times higher than Boma Bungu), and the incidence rate was 2.19 (i.e. 37% higher).

The lower ICER for the strategy with vector control in Mosango may be due to savings the cost of active screening by achieving EOT and thus ending active screening earlier. i.e Sites with better coverage of active screening would be where vector control would be more likely to be cost-effective. Although this is logical, it justifies giving more resources to sites that already have more.

The key question is which of these two parameters, the percentage of the population covered by active screening and incidence rate, is driving the model result? Although the authors conducted several sensitivity analyses for vector control, time horizon, and discount rate, they didn't address program performance. Would vector control be a minimum cost strategy in Boma Bungu and Budjala if they had the active screening rate of Mosango? It's also possible that the incidence rate is driving the results, which could also be demonstrated by a sensitivity analysis.

More generally, there is considerable interest in whether or not vector control programs are a good use of resources as locations approach elimination of a vector-borne disease. With a better understanding of the role of program performance vs incidence the authors could briefly discuss whether or not their results would generalize to other locations.

Specific comments:

Published CEA often present a table with a summary of model parameters. Although the authors provided descriptive summaries of the five health zones in Table 1, and details on the health outcome and cost parameters in supplementary text, I recommend that they reformat and move the summary tables with health outcome and cost parameter (S31-S35) to the main text.

I understand the editors limit the number of tables and figures in the main text, and would propose that Figure 3, which repeats information in Table 4 could be replaced with the summary table of health outcomes and parameters.

Page 2, last paragraph of introduction. Please define the terms "efficiency", "cost-effectiveness," and "comparative efficiency," and use them consistently. For example, in the last sentence, of this paragraph, it's unclear how to define "comparative efficiency" in a way that doesn't involve economic factors. I'm not aware that "comparative efficiency" is a synonym for "cost-effectiveness."

Page 2, results paragraph 2, Please state which treatments are used in the analysis. The authors report the treatments in the Discussion (page 31, line 3), but the statement should be moved from the discussion to this paragraph.

Page 3, Section 2.1, First paragraph. Did the authors select 10,000 model iterations by convention or experiments with alternatives?

Also, is the correct term “reactivation of active screening” or “reactive screening”? Please use correct term consistently.

Table 2. Please clarify the relationship between EOT and end of AS. The text (Page 3) states that AS ends when no cases have been reported for 3 consecutive years. If EOT means no cases, then I would expect AS to end at least 3 years after EOT. In the Budjala mean AS strategy results, explain why EOT and AS occur at the same time.

Table 2. Add the definition of the acronym (RS=reactivation of active screening) and correct the relevant column heading.

Page 6, Section 2.4, paragraph 1, Please add the currency year. i.e. Replace “present day” with the currency year “(20xx US\$)”.

Page 8, Section 2.4.2, first sentence. Clarify that the authors chose not to report uncertainty intervals in their ICER estimates. It is certainly possible to incorporate uncertainty in cost-effectiveness analyses.

Page 9, Discussion paragraph 2. According to Table 3, optimal strategies would not exceed \$1.37 rather than 1.35.

Figure 2. It’s difficult to distinguish pink and yellow in print. Maybe red would be easier than pink to distinguish from yellow.

Page 12, Section 4.1. With a fixed absolute population seen by active screening and 3% population growth, doesn’t the percentage of the population seen by active screening decline over time? Please explain why this parameter is fixed.

Figure 3 is not a standard decision tree. The first node (Stage 1 vs Stage 2 + Stage 1 treatment failures) is not correct, because a person can only move to treatment failure after Stage 1. Similarly, the second node for Stage 2 and Stage 2 treatment failures is incorrect, because a person can only move to treatment failure after Stage 2. A simple correction would be to make the first node Stage 1, Stage 2, and late-stage disease, which would represent the patient’s stage when detected. The authors could note in the figure legend that the Stage 2 node applies to Stage 1 treatment failure, and late-stage disease note applies to Stage 2 treatment failures.

Reviewer #2 (Remarks to the Author)

I appreciate very much this opportunity to review this important manuscript. This study evaluates the cost-effectiveness of eliminating Human African trypanosomiasis from the Democratic Republic of Congo. It assesses the population impact and cost-effectiveness of four scenarios that involved passive screening, active screening and vector control and their combinations. Overall, I have found this study to be very robust, both their data source and the analytic approach they employed. The discussion follows closely from their findings. In particular, the key strength their prediction of the probability of achieving EOT by the year 2030. I agree the paper has important relevance to public health and its audience.

I have a few major concerns:

Major, Reviewer #2 (there is no reviewer #1)	
Comment	Response
1. The major concern is the method of models; in the main text, it has clearly stated that an SEIRS deterministic model has been used. In the appendix, it appears that it is a decision-analytic model was used, and the SEIRS is embedded in the decision tree model. It is unclear how the two models are working together (Figure S2)? In particular, the SEIRS model uses a set of differential equations. This set of equations was not presented anywhere in the main text or the appendix?	We thank the reviewer for the comment and we understand that we were unclear about the link between the SEIRS model and the decision analytic model. We have changed the previous Figure S2 (now Figure S4) to more completely show the relationship between the dynamic (SEIRS) model and the treatment model. We have added the following sentence to the caption to make clear where the fitting of transmission model parameters and simulation of the model are described for the whole of DRC in other publications (Crump et al 2021, Huang et al 2020 under review in the same journal). We have pointed the combination of both models and the existence of Figure S2 in the first line of the methods: We employed a previously published dynamic, transmission "susceptible-exposed-infected-recovered-susceptible (SEIRS)" model, combined with a probability tree of treatment (Figure 4), in a manner fully illustrated in Figure S4).

2. Further, the model structure for each scenario is essentially identical. However, it is unclear how the impact of each of these intervention strategies was assessed? For example, how vector control would change the transmission probability or force of infection in the SEIRS model? it	Thanks for the comments. We hope that with the edit to Figure S4 above that this issue is now a bit clearer. The force of infection is affected by two mechanisms: 1) interrupt the vector or 2) decrease the number of extant infections and both of these mechanisms are governed by the ODE of the SEIRS model. These equations are further elaborated in the companion paper, by Huang et al (2020). The treatment model was identical, yes.
Minor comments	
3. I realise the authors used lots of parameters from Uganda since DRC-specific data may not be available (Appendix S79-S85). Could the authors comment on the accuracies of using data from Uganda?	This was only true for VC. There's upcoming work that shows that the Uganda and DRC parameters for vector control are similar, and the costs differ due to the extent of the operations, so we adapted to the DRC context as detailed in section 6.6.18-20 in pages S89. However, this is not yet published. By comparison, we know that an analysis from Chad published after our analysis was completed showed that the cost per square kilometer in Chad and Uganda were similar (\$1,247 vs \$1,373, respectively). We have added a note to the methods to note this (page 15, second paragraph). For treatment, active screening (screening by mobile teams) and passive screening (screening of symptomatic individuals in fixed health facilities) are gathered from DRC sources, as detailed in sections 6.6.1-17 on pages S80-S85 of the text and cited in references 18, 19, and 26.
4. It would be interesting to use probability sensitivity analysis to investigate how the change of WTP may impact the probability of cost-effectiveness. Could the author include this information (PMID: 32971056, Figure 2). Further, would a different WTP also impact the probability of achieving EOT by 2030?	The cost-effectiveness acceptability curves (the class of graph in Figure 2 of PMID: 32971056) are indeed included in the manuscript, just in the supplement (see Figure S7 and S8, page S42-S43 - renumbering has occurred due to other additions to the supplement). In the interest of space and parsimony, we have excluded this from the main paper, and we have shown the headline results in table 4 and figure 3, alongside more information. We now point this out in the results and the caption to the

	table. The first line of the results section 2.4: The cost-effectiveness of each strategy in each health zone is displayed in Table 4, select features are illustrated in Figure 3, and more traditional cost-effectiveness acceptability frontiers (CEAFs) are shown in Figure S7-S8. The caption of the table: Strategies highlighted in pink are optimal strategies: the strategies for which the mean net monetary benefit (NMB) is highest, equivalent to the information found in cost-effectiveness acceptability frontiers (CEAFs), which are shown Figures S7-S8. To answer the reviewer's question explicitly, yes, a higher WTP would indeed ensure the EOT by 2030, but we found that the WTP needed to be just \$4/DALY in Kwamouth, and the EOT strategy was cost-saving at a 20-year time horizon in Mosango. In the second paragraph of the discussion: In Mosango and Kwamouth the addition of VC to Mean AS is predicted to expedite EOT and be cost-effective for low WTP; while the addition of VC would ensure EOT while saving costs in Mosango, the WTP to choose the strategy that ensures EOT is only \$4/DALY (Tables 2 and 4). However, in Boma Bungu and Budjala current practice appears sufficient to ensure EOT by 2030 despite Boma Bungu's "moderate" incidence and Budjala's low AS coverage (<0.5%) (Tables 1 and 4). In Yasa Bonga, where VC has been coupled with a relatively high coverage rate (57% of the 2016 population) EOT is estimated to have occurred already (Table 2).
--	---

Reviewer #3 (Remarks to the Author)

Major, Reviewer #3 (there is no reviewer #1)

Comment	Response
The authors conducted a cost-effectiveness and net-benefit analysis of four strategies for elimination of Gambiense human African trypanosomiasis (gHAT) in the Democratic Republic of Congo. A strength of their analysis is that it's based on impressive data on the epidemiology of gHAT and health program performance in the five settings in recent years. Their data on actual health programs also raises a concern about perpetuating inequity among the sites. The results favor current practice (passive screening and mean active screening) in 2 of 4 settings where vector control is not currently implemented. In those 2 settings, Boma Bungu and Budjala, the percentage of the population covered by active screening is relatively low (7.2% and 0.41% respectively), and incidence is low (1.37 and 0.05 per 10,000 population respectively). The main differences between Mosango, where adding vector control is cost effective, and Boma Bungu, are that the percentage of the population with active screening in 2016 was 34% in Mosango (i.e. 5 times higher than Boma Bungu), and the incidence rate was 2.19 (i.e. 37% higher). The lower ICER for the strategy with vector control in Mosango may be due to savings the cost of active screening by achieving EOT and thus ending active screening earlier. i.e Sites with better coverage of active screening would be where vector control would be more likely to be cost-effective. Although this is logical, it justifies giving more resources to sites that already have more.	We thank the reviewer for the thoughtful review of our paper. The issue of inequity and varying population screening is indeed an important issue. If we were only counting the reported incidence in our DALY calculations, then indeed we would be reinforcing inequalities in service provision. However, a substantial portion of the team's work has been in examining the degree to which one can infer underlying incidence given variations in population screening, as well as the positivity rate among those screened; that analysis is detailed in Crump et al 2021 (PLOS Computational Biology). Further, that analysis not only addresses the assessment of underlying transmission, but also the deaths outside of healthcare. In fact, DALYs are the combination of both cases and deaths (primarily deaths outside of care, since treatment is highly successful when accessed), and therefore it is not the 'apparent' (reported) incidence that is driving the DALYs, but the estimated underlying deaths, which increase when there is less screening (and thus, less treatment). Therefore, if screening is low and the positivity rate is high, the decision analysis would favor investing more resources to increase screening, but since screening seems to be low as well as the positivity rate, then the analysis does not indicate that any increase in activities would be efficient. In that paper (Crump et al) one can see that the low coverage/low positivity of the western provinces indicates low overall incidence, but the low coverage/high positivity of the eastern provinces indicates that there is high overall incidence. We are working on CEAs for those provinces, but that's where enhanced resources would be most optimally used. We decided to run Strategy 1 (Mean AS, alongside standard PS included in all

The key question is which of these two parameters, the percentage of the population covered by active screening and incidence rate, is driving the model result? Although the authors conducted several sensitivity analyses for vector control, time horizon, and discount rate, they didn't address program performance. Would vector control be a minimum cost strategy in Boma Bungu and Budjala if they had the active screening rate of Mosango? It's also possible that the incidence rate is driving the results, which could also be demonstrated by a sensitivity analysis.

More generally, there is considerable interest in whether or not vector control programs are a good use of resources as locations approach elimination of a vector-borne disease. With a better understanding of the role of program performance vs incidence the authors could briefly discuss whether or not their results would generalize to other locations.

strategies) because it was helpful for stakeholders we spoke to understand the status quo (current) investment of resources, how likely the status quo was to reach EOT, how the status quo resources may change in the future if it can reach very low incidence or EOT and cessation of activities. Then, the Max AS option would bring up the coverage of those low-coverage health zones to more than 30% of the population, thus letting us think about the return on extra investments. The simulations now show that such activities would avert some deaths, in and out of care, though not many (certainly not be 3x-30x as many, which would be the concomitant increase in coverage) (see table 4). This indicates that there would be strong diminishing returns to enhanced investments in those low-incidence health zones (Boma Bungu and Budjala). Therefore, at coverage levels similar to Mosango, no, the strategy to include VC is not optimal.

In short, it is true that operational capacity (in terms of active screening coverage) declines as incidence falls, but by back-calculating the incidence taking into account both screening and positivity rate, we arrive at an estimate of the true incidence and we do not believe that we are reinforcing inequities.

To make things clearer, we have edited two portions of the results. In the first paragraph of the results (page 2):

The model was fitted to case data from the health zones, adjusting for both the screening coverage as well as the positivity rate, and projections for reported cases, unreported illnesses, and reported and unreported deaths were simulated for 2020--2050 (Crump et. al 2021, Huang et. al 2020).

On the second paragraph of the results (page 2):

Status quo strategies, or the current practice, are considered to be equivalent to Strategy 1, except in Yasa Bonga, where VC has been in place since mid-2015, and therefore we

	consider the status quo equivalent to Strategy 3 and we omit strategies without VC for Yasa Bonga. The strategies with historical maximum screening coverage allow us to compare the outcomes in all health zones when high amounts of resources are invested while still taking into account that maximum capacity might vary across locations.
Minor comments	
Published CEA often present a table with a summary of model parameters. Although the authors provided descriptive summaries of the five health zones in Table 1, and details on the health outcome and cost parameters in supplementary text, I recommend that they reformat and move the summary tables with health outcome and cost parameter (S31-S35) to the main text. I understand the editors limit the number of tables and figures in the main text, and would propose that Figure 3, which repeats information in Table 4 could be replaced with the summary table of health outcomes and parameters.	Model parameters are described in tables S31-S35 and sections S6 in pages S54-S81 in great detail in the supplement. To make it easier to quickly look up the values of parameters, we now provide a condensed (and long) version of the tables S31-S35 with the names, distributions, and mean & 95% CIs, starting in page 13 of the manuscript (in the methods section). In the caption, we note that the parameters are available in greater detail and with sources in the parameter glossary of the supplement.
Page 2, last paragraph of introduction. Please define the terms “efficiency”, “cost-effectiveness,” and “comparative efficiency,” and use them consistently. For example, in the last sentence, of this paragraph, it’s unclear how to define “comparative efficiency” in a way that doesn’t involve economic factors. I’m not aware that “comparative efficiency” is a synonym for “cost-effectiveness.”	We’ve changed the term “comparative efficiency” to “cost-effectiveness” to avoid confusion.
Page 2, results paragraph 2, Please state which treatments are used in the analysis. The authors report the treatments in the Discussion (page 31, line 3), but the statement should be moved from the discussion to this paragraph.	Good point. We have added this line to the end of the second paragraph of the results: Treatment is modelled by a branching (probability tree) model simulating WHO-recommended treatment, which consists of fexinidazole on an inpatient or outpatient basis, and pentamidine or nifurtimox-

	eflonithine combination therapy (NECT) when fexinidazole is contraindicated based on patient characteristics. (see section S1.2.3 for a breakdown of treatment groups)\cite{WHO2019guidelines}.
Page 3, Section 2.1, First paragraph. Did the authors select 10,000 model iterations by convention or experiments with alternatives?	We've now added trace plots of DALYs and costs, as well as cumulative means, cumulative lower PI (2.5th percentile), and cumulative higher PI (97.5th percentile (Figure S2 and S3). The most important metrics of the distribution (the mean and the predictive intervals) seem to be unchanged with the addition of the last 2000 iterations. Therefore, we believe that 10,000 iterations are enough. Then we added some text to the end of section S1.2.1: To characterise the uncertainty throughout the model, we used 10,000 draws from the posterior of the transmission model. Trace plots of the year of elimination of transmission (EOT) and probability of reactive screening (RAS) are displayed in Figures S2-S3, indicating that the features of the distribution are stably estimated with 10,000 samples. We also know from experience that if one selects to run the model with too few iterations, the cost-effectiveness acceptability curves will not look smooth, whereas Figures S5 and S6 look smooth, so therefore, we believe that 10,000 iterations were enough. Furthermore, upon repeated re-executions of the code our headline results have been consistent.
Also, is the correct term “reactivation of active screening” or “reactive screening”? Please use correct term consistently.	It's reactive screening. Thanks for pointing this out. We've removed "reactivation of active screening".
Table 2. Please clarify the relationship between EOT and end of AS. The text (Page 3) states that AS ends when no cases have been reported for 3 consecutive years. If EOT means no cases, then I would expect AS to	EOT does not mean no cases reported, and no cases does not mean EOT:  1) EOT may occur, but due to the long natural history of gambiense trypanosomiasis, people may be

end at least 3 years after EOT. In the Budjala mean AS strategy results, explain why EOT and AS occur at the same time.	reported ill years after they've been infected, before transmission chains were broken. 2) Although no cases may be reported for a period of time, imperfect activities mean that transmission may remain, so we do not call the absence of cases EOT, hence the continued presence of passive screening in fixed health facilities and the potential for re-active screening. The dynamics above are evidenced in Table 2. Situation #1 above may occur using strategies that include VC implementation, when it is probable that EOT occurs even before cases stop being reported, as the absence of tsetse would break transmission chains although the human reservoir (infected individuals) remains. Situation #2 above is also shown in table 1, where in 1-62% of our model iterations reactive surveillance must be deployed to address the newly found remaining infection in the community. As EOT is a latent characteristic, and it has been assessed by other modeling work that 3 years of no cases is a relatively safe policy for cessation of activities, and is in line with the WHO recommendation for cessation of activities. This indicates that either EOT has occurred or that R_e (R-effective) is under 1, and the transmission dynamics are such that the EOT is imminent (Castaño et. al JID 2019). However, in order to stop AS activities in practice, the national programme need a measurable metric, which is the number of cases reported.
Table 2. Add the definition of the acronym (RS=reactivation of active screening) and correct the relevant column heading.	Thanks for the suggestion. The caption for table 2 has not been amended.
Page 6, Section 2.4, paragraph 1, Please add the currency year. i.e. Replace “present day” with the currency year “(20xx US\$)”.	Thank you for this suggestion. We've edited the last sentence of the first paragraph of 2.4. (2018 US\$).

Page 8, Section 2.4.2, first sentence. Clarify that the authors chose not to report uncertainty intervals in their ICER estimates. It is certainly possible to incorporate uncertainty in cost-effectiveness analyses.	We agree that it is possible to incorporate uncertainty in cost-effectiveness analyses, and this is the reason for our use of the net benefits framework, and presentation of the CEAFs in figure S7 and S8 and the uncertainty at each WTP in table 4. However, CIs for ICERs when there are more than 2 intervention are invalid. Since the ICER is *incremental* to the next best strategy (not to the comparator), in each iteration, there may be strongly and weakly dominated strategies that would vary from iteration to iteration, and the *incremental* value would become unclear when the comparator of the *next best* strategy is unclear overall. For this, linearizing ICERs in the way that NMB does overcomes the issue of weakly and strongly dominated strategies (see Briggs 2006 book, Decision modeling for health economic evaluation, chapters 4 and 5). Second, CIs for ICERs would be invalid since the negative ICERs could be indicative of cost-savings or additional cases incurred (negative ratios could mean either) but in terms of policies, these scenarios are not interchangeable. This issue has been discussed in the literature (see Andrew Briggs 2000, Pharmacoeconomics, 17(5), 479-500). Lastly, it is actually recommended that the mean ICER are not consistent with first principles of efficiency. Rather it should be the ratio of the mean DALYs and mean costs that ought to be used (for more, see Stinnett & Paltiel 1997, Med Decis Making 1997;17:483-489).
Page 9, Discussion paragraph 2. According to Table 3, optimal strategies would not exceed \$1.37 rather than 1.35.	Thank you for noticing this. The text has been duly amended.
Figure 2. It's difficult to distinguish pink and yellow in print. Maybe red would be easier than pink to distinguish from yellow.	We appreciate the suggestion; indeed the categories would be a bit difficult to distinguish. Because we chose a color scheme that would match across different figures (those colors are also used on the

	web interface and other project materials), we chose instead to separate the categories by thin white lines in Figure 2, rather than reconfiguring the color palette. We hope this makes the figure easier to interpret.
Page 12, Section 4.1. With a fixed absolute population seen by active screening and 3% population growth, doesn't the percentage of the population seen by active screening decline over time? Please explain why this parameter is fixed.	Either decision would have been fine, but this way we keep the economic expenditure of AS and PS constant over time. Anyway, the effects of this policy already show that disease is on the decline, therefore obviating the need for additional resources allocated to scale with population growth. The decision on whether to grow the economic resources allocated would be an additional question that was not expressed by partners in DRC.
Figure 3 is not a standard decision tree. The first node (Stage 1 vs Stage 2 + Stage 1 treatment failures) is not correct, because a person can only move to treatment failure after Stage 1. Similarly, the second node for Stage 2 and Stage 2 treatment failures is incorrect, because a person can only move to treatment failure after Stage 2. A simple correction would be to make the first node Stage 1, Stage 2, and late-stage disease, which would represent the patient's stage when detected. The authors could note in the figure legend that the Stage 2 node applies to Stage 1 treatment failure, and late-stage disease note applies to Stage 2 treatment failures.	We thank the reviewer for their suggestion, and we have changed Figure 3 accordingly. We then added the following text to the caption: Stage 2 disease treatment sometimes applies to stage 1 treatment failures, and some late-stage disease includes some patients who were stage 2 treatment failures. The outcomes of the small proportion of cases that experience unsuccessful treatment are determined by calculating the product of the probability of unsuccessful treatment and the outcome of disease at a later stage of disease. Cases are assumed to go through treatment at most twice. We have additionally illustrated the connection between the probability tree of treatment and the dynamic model of transmission in Figure S2, at the request of reviewer 2.

Old and new figure 3:

REVIEWERS' COMMENTS

Reviewer #2 (Remarks to the Author):

I appreciate very much again for the opportunity to see a revision of this article. I have gone through the paper in detail again, and I appreciate the changes the authors have made. I think they have addressed my previous concerns to satisfaction. I am also happy with the presentation of Figure S4, which demonstrate the dynamics model and decision tree model very well.

However, I would like to recommend the authors to include the CHEERS check for cost-effectiveness analysis as an appendix, as this has become a standard in the field of health economic evaluation.

Reviewer #3 (Remarks to the Author):

Congratulations on your very good research and a well-written manuscript. No further comments.